# LOVE-R1: Advancing LOng VidEo Understanding with An Adaptive Zoom-in Mechanism via Multi-Step Reasoning

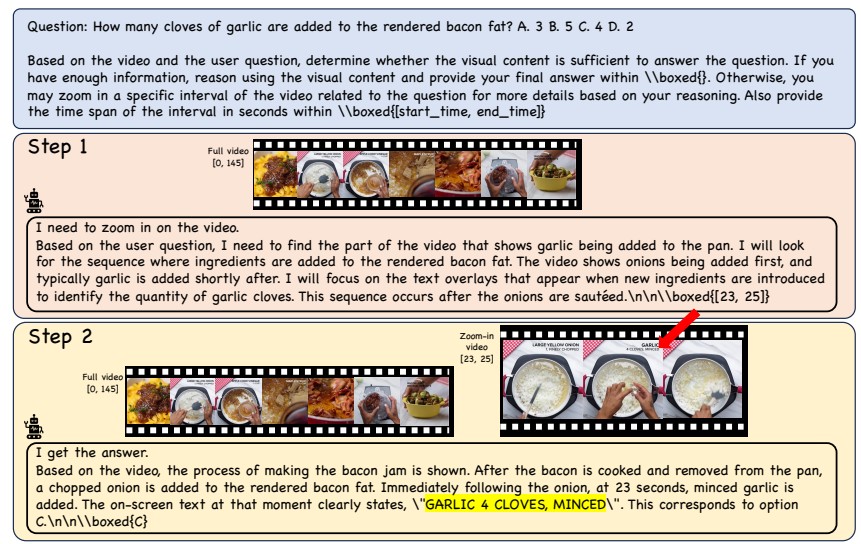

Figure 1: Illustration of the workflow of LOVE-R1. Our model first takes densely sampled small-resolution frames from the whole video as inputs to understand the video globally. If needed, it can adaptively zoom in on a video clip to gain fine-grained spatial details. The workflow is implemented as a multi-step reasoning process.

## ABSTRACT

Long video understanding is still challenging for recent Large Video-Language Models (LVLMs) due to the conflict between long-form temporal understanding and detailed spatial perception. LVLMs with a uniform frame sampling mechanism, which samples frames with an equal frame size and fixed sampling rate, inevitably sacrifice either temporal clues or spatial details, resulting in suboptimal solutions. To mitigate this dilemma, we propose LOVE-R1, a model that can adaptively zoom in on a video clip. The model is first provided with densely sampled frames but in a small resolution. If some spatial details are needed, the model can zoom in on a clip of interest with a large frame resolution based on its reasoning until key visual information is obtained. The whole process is implemented as a multi-step reasoning process. To train the reasoning ability, we first finetune the model on our collected 38k high-quality CoT data and enhance it with decoupled reinforcement finetuning. As outcome rewards can not provide fine-grained process supervision, we decouple multi-step reasoning into multiple single-step reasoning and optimize the internal zoom-in ability explicitly. Experiments on long video understanding benchmarks show that our model with the slow-fast adaptive frame sampling mechanism achieves a great trade-off between sampling density and frame resolutions, and LOVE-R1 outperforms our baseline Qwen2.5-VL by an average of 3.1% points across 4 common long video understanding benchmarks.

# 1 INTRODUCTION

Large Video-Language Models (LVLMs) (Zhang et al., 2024b; Bai et al., 2025; Zhang et al., 2024c; Fu et al., 2025b) have achieved great progress in understanding temporal dynamics. However, long video understanding (LVU), owing to the long-form temporal dependency and the great variety of action sequences, still poses great challenges to them. When tackling long videos, mainstream LVLMs utilize a uniform sampling strategy, in which frames are sampled with a fixed interval and resolution. Constrained by the context length, LVLMs will face the dilemma of balancing spatial resolution and temporal sampling density. Sampling more frames can help better understand motion clues, while adopting a larger frame resolution can preserve more spatial details. With a limited context length, LVLMs with a fixed sampling strategy fail to balance the spatial-temporal trade-off.

However, only a small number of keyframes are needed for a large proportion of the questions. Suo et al. (2025) find that recent LVLMs can achieve more than 75% Pass@N accuracy with 32 randomly sampled frames on most long video understanding benchmarks when $N$ is larger than 40. Furthermore, ViLAMP (Cheng et al., 2025) also finds that around 90% of query-induced attention weights concentrate only on 5% of frames. These findings show that selecting high-quality keyframes is crucial for effective and efficient long video understanding.

Inspired by the strong reasoning capacity demonstrated by recent reasoning models (Guo et al., 2025; Jaech et al., 2024), our objective is to train an LVLM with adaptive zoom-in ability. Specifically, three abilities are needed: 1) the model with the **decision ability** can decide whether the visual information is sufficient to answer the question, 2) if not, LVLMs can use the **zoom-in ability** to select the most relevant time span to zoom in, 3) when visual information is sufficient, LVLMs use the **answering ability** to provide answers through thinking. This adaptive frame selection mechanism allows LVLMs to attend to informative frames with a large resolution while understanding the overall event with a small resolution, thus preserving vision tokens within a manageable context and balancing long-form temporal understanding and detailed spatial perception.

Based on this motivation, we propose LOVE-R1, a long video understanding model with a slow-fast-like dynamic frame processing mechanism, as shown in Figure 1. We first sample the video at a high frame rate (e.g., 768 frames) but in a small resolution (e.g., 32 tokens per frame) to provide the model with a global view of the video without sacrificing temporal details. When some spatial details are needed, we provide the model with a few high-resolution frames (e.g., 256 tokens per frame). The entire process is automatically decided by the model itself. The development of LOVE-R1 undergoes a three-stage post-training: 1) **slow-fast template finetuning**: Instead of adopting a fixed frame sampling strategy, LOVE-R1 processes the video into multiple segments with different frame rates, resolutions and timespans. We finetune the LVLMs with open-sourced video instruction data to adapt them to the new video template. 2) **CoT cold start**: We construct 38k CoT data with careful data selection, construction, cleaning, and filtering. After finetuning on the high-quality CoT data, the model is equipped with basic reasoning ability. 3) **decoupled reinforcement finetuning**: Reinforcement learning has been shown to be an effective method to boost reasoning capacity. However, most of the methods are based on outcome rewards, i.e., the final answer is correct or not, which can not provide fine-grained process rewards in our multi-turn scenario. Thus, we decouple the multi-turn conversations into multiple single-turn conversations and optimize the zoom-in ability separately, which is the key factor for long video understanding and can not be optimized effectively in the standard GRPO algorithm (Guo et al., 2025).

With the ability to zoom in on the video, LOVE-R1 achieves state-of-the-art performance on common long video understanding benchmarks. Specifically, LOVE-R1 gets 48.2% on LVBench (Wang et al., 2025b), 60.1% on LongVideoBench (Wu et al., 2024), and 66.2% on VideoMME (Fu et al., 2025a), outperforming our baseline Qwen2.5-VL 7B (Bai et al., 2025) by 6.2%, 4.1%, and 1.0%. We hope our work can provide a new paradigm to tackle the long video understanding problem.

# 2 RELATED WORK

## 2.1 LONG VIDEO UNDERSTANDING WITH LARGE VIDEO-LANGUAGE MODELS

In order to unify image and video representation and pretraining, recent Large Video Language Models (LVLMs) (Bai et al., 2025; Zhao et al., 2025b; Peng et al., 2025; Zhang et al., 2024c) adopt

a uniform sampling strategy, in which frames are sampled from the video with a fixed interval and resolution and frames are concatenated in order. Although this fixed dense sampling strategy is simple and effective in the short video scenario, the number of visual tokens will soon increase out of the budget when tackling long videos.

To preserve informative visual information and reduce visual tokens, a lot of video processing methods are proposed: 1) For **token compression methods** (Li et al., 2024d; Song et al., 2024; Shen et al., 2024; Man et al., 2025; Shu et al., 2025), they prune or merge visual tokens based on similarity or relation to the query. Tokens after pruning are poorly organized. 2) For **keyframe selection methods** (Zhang et al., 2025b; Wang et al., 2025a; Hu et al., 2025), they prune visual information at a larger granularity. The most informative frames are selected for inference. Since the structure of frames is not changed, these methods can serve as a plug-and-play module to other LVLMs without finetuning. 3) For **long context methods** (Chen et al., 2024b; Shen et al., 2025; Ren et al., 2025), they extend the context window to preserve as much information as possible. 4) For **agent-based methods** (Luo et al., 2024; Liu et al., 2025b; Wang et al., 2025c), they use tools to handle sub-videos separately and then use an LLM to select and merge the helpful information and then answer the question. In this work, we propose a dynamic video processing method that can zoom in on a subset of the video adaptively, striking a balance between sampling density and frame resolution.

## 2.2 MULTIMODAL REASONING

Recent reasoning models Guo et al. (2025); Jaech et al. (2024) show that generating long Chain-of-Thoughts (CoT) at test time, which breaks a hard problem into a series of solvable sub-problems and then derives the final answer, can significantly enhance the performance. Early exploration of video reasoning (Feng et al., 2025; Yang et al., 2025; Chen et al., 2025b; Zhao et al., 2025a) also shows that taking visual information into thinking can boost performance in both perception and reasoning tasks. Different from reasoning on pure text, multimodal reasoning has greater flexibility to manipulate visual content to assist reasoning, such as zooming in on the image region (Su et al., 2025; Zhu et al., 2025) and grounding related objects (Zhang et al., 2025c; Fan et al., 2025). In this work, we aim to use the reasoning ability to decide which video clip to zoom in on so that models can process long videos in a limited context.

Very recently, two concurrent works VITAL (Zhang et al., 2025a) and Video-MTR (Xie et al., 2025) also share the idea of dynamically processing video information via CoTs. Differently, we adopt a slow-fast video template rather than interleaving video clips with CoTs, which can balance the temporal density and frame resolution while preserving pretraining performance as much as possible. Further, we propose to provide fine-grained process rewards by decoupled reinforcement finetuning, achieving higher performance.

## 3 A DYNAMIC FRAME PROCESSING MECHANISM

Previously, Large Video Language Models (LVLMs) usually adopt a uniform frame sampling mechanism, in which frames are sampled at a fixed sampling rate, and frames are of equal size. However, as the video becomes longer, the number of visual tokens will inevitably exceed the context length. Both decreasing the sampling rate or frame resolution can reduce visual tokens but at the cost of losing temporal or spatial details, failing to understand the long video fully and accurately.

To enable better temporal-spatial perception with a constrained context length, we propose using a slow-fast-like template as shown in Figure 2(c). Specifically, for a video with $T$ seconds, we first obtain the fast video by densely sampling the video at a high frame rate $fps^f$ but in a small resolution $r^f$. The total number of sampled frames $N^f$ does not exceed a predefined maximum frame number $N_{max}^f$, i.e. $N^f = \min\left(T \times fps^t, N_{max}^t\right)$. This fast video provides the overall event of the video with rich temporal details. When the model needs some spatial details from a specific clip $[t_1, t_2]$, we sample the frames within this clip at a high resolution $r^s$ and a small number of frames $N_{max}^s$. The slow video is encoded separately and appended after the fast video. If the model zooms in on multiple clips, the slow videos will be organized in order. The overall video template is as follows:

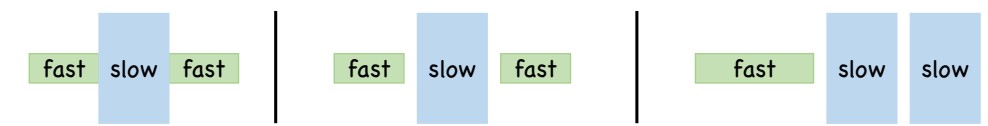

(a) unified video template     (b) interleaved video template     (c) appended video template (Ours)

Figure 2: Different slow-fast video templates. Templates (a) and (b) will replace the original fast video segments with the slow videos. Template (a) treats multiple video segments as a whole video while Template (b) explicitly separates them with identifiers ($<$|vision_start|$>$, $<$|vision_end|$>$). Template (c) appends the additional slow videos at the end of the fast video without removing the corresponding fast video segments. We adopt Template (c).

---

Full video [0,T]: $\langle$fast_video$\rangle$ Subset zoom-in video clip $[t_1, t_2]$: $\langle$slow_video_1$\rangle$ Subset zoom-in video clip $[t_3, t_4]$: $\langle$slow_video_2$\rangle$ ... Subset zoom-in video clip $[t_{2k-1}, t_{2k}]$: $\langle$slow_video_k$\rangle$

---

where $t_1 \le t_3 \le \cdots \le t_{2k-1}$ and $\langle \cdot \rangle$ will be replaced with video tokens.

We also ablate other video templates as shown in Figure 2 (a) and (b). These two templates will replace the corresponding fast video clips with slow videos. Template (a) views multiple segments as a whole video, while Template (b) separates them explicitly. These two templates will break the fast video into multiple videos. Instead, our template (c) appends the slow videos behind the fast video. We find this template aligns well with the pretraining template, thus it can be adopted by the model with a little finetuning data. With the dynamic frame processing mechanism, the model strikes a balance between temporal sampling density and spatial resolution.

However, which video clip is necessary to zoom in on is related to the user query and not easy to find. Thus, we aim to use the strong reasoning capacity of LLMs to determine it so that the model needs three reasoning capacities: the decision ability, the zoom-in ability, and the answering ability. Given a specific query, the model should first decide whether the visual information is sufficient to answer the question (the decision ability). If not, the model can use the zoom-in ability to select the most relevant time span to zoom in. When visual information is sufficient, LVLMs use the answering ability to provide the answer through deep thinking. The overall pipeline works in a multi-step manner. In the following, we show how to build a base model to a reasoning one via a three-stage post-training.

## 4 A THREE-STAGE POST-TRAINING RECIPE

### 4.1 STAGE 1: SLOW-FAST TEMPLATE FINETUNING

In this work, the proposed dynamic zoom-in mechanism adopts a new video template and requires strong temporal awareness to precisely localize the relevant video clip. The videos in our slow-fast template will be represented in multiple segments with different frame rates, resolutions, and timespans, which is different from the pretraining one. Thus, we conduct an extra supervised fine-tuning stage before cold start to maintain the video understanding ability under the new template and enhance the temporal grounding ability. Specifically, we use FineVideo (Farré et al., 2024) and videos ranging from 2 to 3 minutes in LLaVA-Video-178k (Zhang et al., 2024c) as the general video instruction tuning dataset to enhance the perception ability and ET-Instruct (Liu et al., 2024) as the temporal grounding dataset to strengthen the temporal grounding ability. The total number of training data is around 153k. During training, we use the ground truth timespans or random timespans to obtain several slow videos to simulate the slow-fast template. After finetuning, the model successfully adopts the new template without sacrificing performance.

### 4.2 STAGE 2: CoT COLD START

To facilitate training, we annotate 38k CoT data with strong proprietary reasoning models (Comanici et al., 2025; Hurst et al., 2024). The source videos are selected from two widely used grounded video question answering datasets NExT-GQA (Xiao et al., 2024) and CG-Bench (Chen et al., 2025a).

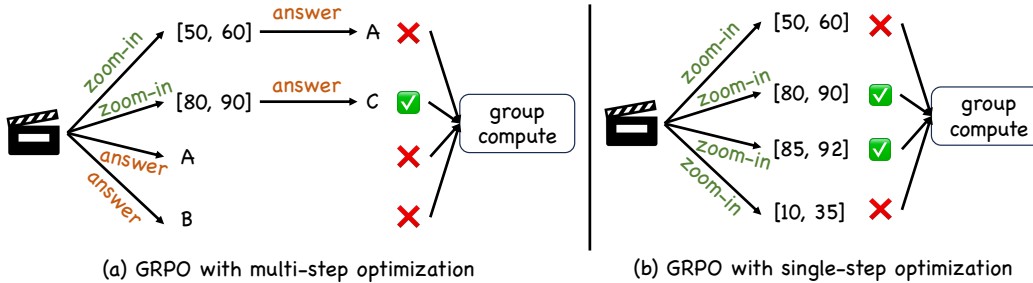

(a) GRPO with multi-step optimization        (b) GRPO with single-step optimization

Figure 3: Illustration of decoupled reinforcement finetuning. (a) For questions without ground truth timespans, we apply the standard GRPO algorithm to optimize multi-step CoTs as a whole. (b) To provide fine-grained process rewards, we decouple multi-step reasoning into multiple single-step reasoning and optimize the single zoom-in step explicitly by appending the zoom-in prefix.

Each question in these datasets is annotated with related timespans so that we can filter out CoTs with wrong zoom-in timespans. The entire construction pipeline undergoes strict cleaning, filtering, and prompt engineering to ensure data quality. The detailed construction pipeline can be found in Appendix B. And an example of our collected CoT is shown in Figure 1.

With the 38k CoT data generated above, we finetune the base model on it to learn the decision ability, zoom-in ability, and answering ability. We treat each single step as a sample and train the model on each single step. To ease learning difficulty and make flexible control during training and inference, we explicitly decouple the three abilities by adding a prefix for each CoT. We add "I get the answer." for answer CoTs and "I need to zoom in on the video." for zoom-in CoTs. With the prefixes, we can precisely control the model behavior during training and testing by adding the corresponding prefix before generating CoTs. Further, to simulate the real-world decision scenarios, we select different slow videos during training. For zoom-in CoTs, we select no slow videos or wrong slow videos for training. For answer CoTs, we select slow videos containing ground truth timespans. Training with different slow videos for different CoTs, models can learn to zoom in on a video clip when the information is not enough and provide a final answer when key visual clues are gained.

### 4.3 STAGE 3: DECOUPLED REINFORCEMENT FINETUNING

Although reinforcement algorithms based on rule-based outcome rewards, like GRPO (Guo et al., 2025), have shown great effectiveness and scalability, the sparsity of rewards can not provide accurate process supervision, which may hinder the performance, especially in our multi-step scenario. When the model zooms in on an error video clip but gets a correct answer, the outcome rewards will still encourage this undesired behavior, preventing the model to learn the grounding ability. It will not be a rare case since the model with only the fast video as input can also get a high performance, as shown in Table 2c. With wrong slow videos, models may learn to answer the question based solely on the fast video while discarding the slow ones, which will greatly degrade the performance. To this end, in addition to the standard GRPO process which optimizes the multi-step CoT as a whole, we propose to decouple the multi-step problem into multiple single-step reasoning and optimize the single zoom-in step with ground truth timespans.

Specifically, two types of data are used for training: 1) local question data, each of which is annotated with ground truth timespans. 2) general question data, which are abundant in the community and have different question types. We use questions in CG-Bench (Chen et al., 2025a) for the first one while using MovieChat (Song et al., 2024) annotated by VideoMarathon (Lin et al., 2025) for the latter one. As shown in Figure 3(a), for general question data, we use the standard GRPO algorithm for optimization. For each question, we generate multi-step CoTs until we get the final answer $ans$. The accuracy reward $r_a^a$ is determined by whether the predicted answer is correct or not:

$$r_a^a = \begin{cases} 1, & \text{if } ans \text{ is correct,} \\ 0, & \text{otherwise,} \end{cases} \tag{1}$$

The accuracy reward is shared by all steps in the CoT. As shown in Figure 3(b), for data with ground truth timespans, we explicitly optimize the single-step zoom-in ability by appending the prefix "I

Table 1: Evaluation results on long video understanding benchmarks. *: Reproduced by us using 16k context. Our LOVE-R1 is also evaluated with around 16k context. Since LOVE-R1 (stage 1) does not have the ability to zoom in, we uniformly sample 32 frames as the slow video.

| Models | Size | #Frames | Context | MLVU | VideoMME (w/o sub) | | LongVideoBench | LVBench |
|---|---|---|---|---|---|---|---|---|
| | | | | | Overall | Long | | |
| Duration | | | | 3~120 min | 1~60 min | 30~60 min | 0~60 min | 4101 sec |
| *Proprietary Models* | | | | | | | | |
| GPT4-V | - | 1fps | - | - | 60.7 | 56.9 | - | - |
| GPT4-o | - | 1fps | - | 66.2 | 77.2 | 72.1 | 66.7 | 34.7 |
| *Open-Source Video MLLMs* | | | | | | | | |
| Video-LLaVA (Lin et al., 2024) | 7B | 8 | - | 47.3 | 40.4 | 38.1 | 39.1 | - |
| LLaMA-VID (Li et al., 2024d) | 7B | 1fps | - | 33.2 | - | - | - | 23.9 |
| ShareGPT4Video (Chen et al., 2024a) | 8B | 16 | - | 46.4 | 43.6 | 37.9 | 39.7 | - |
| LLaVA-NeXT-Video (Zhang et al., 2024b) | 7B | 32 | - | - | 46.5 | - | 43.5 | - |
| VideoLLaMA2 (Cheng et al., 2024) | 7B | 32 | - | 48.5 | 46.6 | 43.8 | - | - |
| LongVA (Zhang et al., 2024a) | 7B | 128 | - | 56.3 | 54.3 | 47.6 | - | - |
| VideoChat2 (Li et al., 2024b) | 7B | 16 | - | 47.9 | 54.6 | 39.2 | - | - |
| LLaVA-OneVision (Li et al., 2024a) | 7B | 32 | - | 64.7 | 58.2 | 46.7 | - | - |
| Vamba (Ren et al., 2025) | 10B | 1024 | - | 65.9 | 57.8 | - | 55.9 | 42.1 |
| VideoChat-T (Zeng et al., 2025) | 7B | 12 | - | - | 46.3 | 41.9 | - | - |
| Quicksviewer (Qi et al., 2025) | 7B | 1fps | - | 61.5 | 56.9 | - | - | - |
| Video-XL (Shu et al., 2025) | 7B | 256 | - | 64.9 | 55.5 | - | 50.7 | - |
| Video-XL-Pro (Liu et al., 2025a) | 7B | 240 | - | 70.6 | 60.0 | - | 56.7 | - |
| LongVILA (Chen et al., 2024b) | 7B | 256 | - | - | 60.1 | 53.0 | 57.1 | - |
| LongVU (Shen et al., 2024) | 7B | 1fps | - | 65.4 | 60.6 | 59.5 | - | - |
| Hour-LLaVA (Lin et al., 2025) | 7B | 1fps | - | - | 63.6 | 55.0 | 60.4 | 45.6 |
| LongVITA-128k (Shen et al., 2025) | 14B | 256 | - | - | 66.4 | 58.8 | 60.9 | - |
| VILAMP (Cheng et al., 2025) | 7B | 1fps | - | 72.6 | 67.5 | 57.8 | 61.2 | 45.2 |
| VideoChat-Flash (Li et al., 2024c) | 7B | 512 | - | 74.7 | 65.3 | 55.4 | - | 48.2 |
| *Open-Source Agent Video MLLMs* | | | | | | | | |
| VideoMind (Liu et al., 2025b) | 7B | - | - | 64.4 | 58.2 | 49.2 | 56.3 | 40.8 |
| Video-RAG (Luo et al., 2024) | 7B | - | - | 72.4 | 62.1 | 59.8 | 58.7 | - |
| *Open-Source Reasoning Video MLLMs* | | | | | | | | |
| Video-MTR (Xie et al., 2025) | 7B | 32 | 4k | 48.4 | 59.0 | 51.0 | - | - |
| Video-R1 (Feng et al., 2025) | 7B | 32 | 4k | - | 59.3 | - | - | - |
| VITAL (Zhang et al., 2025a) | 7B | 1024 | 32k | - | 64.1 | 54.0 | - | - |
| LongVILA-R1 (Chen et al., 2025b) | 7B | 512 | 132k | - | 65.1 | 55.2 | 58.0 | - |
| *Ours* | | | | | | | | |
| Qwen2.5-VL* (Bai et al., 2025) | 7B | 128 | 16k | 66.4 | 65.2 | 54.6 | 56.0 | 42.0 |
| LOVE-R1 (stage 1) | 7B | 768+32 | 16k | 68.5 | 65.4 | 56.0 | 55.6 | 44.7 |
| LOVE-R1 (stage 2) | 7B | 768+32 | 16k | 66.7 | 64.9 | 53.3 | 59.7 | 46.2 |
| LOVE-R1 | 7B | 768+32 | 16k | 67.4 | 66.2 | 57.7 | 60.1 | 48.2 |
| | | | | (+1.0) | (+1.0) | (+3.1) | (+4.1) | (+6.2) |

need to zoom in on the video." before generating the CoT. The accuracy reward $r_a^z$ is determined by whether the predicted zoom-in timespans $[T_1^{pred}, T_2^{pred}]$ overlaps with ground truth timespans $[T_1^{gt}, T_2^{gt}]$:

$$r_a^z = \begin{cases} 1, & \text{if } \text{IoU}([T_1^{pred}, T_2^{pred}], [T_1^{gt}, T_2^{gt}]) > 0, \\ 0, & \text{otherwise,} \end{cases} \quad (2)$$

To align with the accuracy reward for general data and make it distinguishable, we use a binary reward for $r_a^z$ instead of the IoU. After finetuning, the model has the ability to zoom in on a clip of interest from a long video.

## 5 EXPERIMENT

### 5.1 EXPERIMENT SETUPS

**Implementation Details.** Our model is finetuned from Qwen2.5-VL 7B (Bai et al., 2025) with our three-stage training recipe. To enhance temporal grounding ability, we add the frame number in the frames following NumPro (Wu et al., 2025; Ge et al., 2025). For the fast video, we sample at most 768 frames per video, each of which is encoded to 32 tokens (around 168*168 pixels). For each slow video, we sample at most 32 frames, each of which is encoded to 256 tokens (around 448*448 pixels). Due to memory constraints, we set the maximum number of steps to 3, which is around 16k context. During RL training, we mask the whole CoT if the model can not get a final answer at the last step. During inference, the model can adaptively choose whether to zoom in on a video clip or provide an answer before reaching the maximum number of reasoning steps. Upon reaching the final step, we prepend the prefix "I get the answer." to prompt the model to output the

Table 2: Ablation studies. All experiments are tested on Video-MME (w/o subtitle).

(a) Ablation studies on different video templates. Models are finetuned with a part of data.

| Template | Setting | Overall | Long | Medium | Short |
|---|---|---|---|---|---|
| Template (a) | zero-shot | 64.0 | 53.4 | 66.1 | 72.6 |
| | finetune | 63.7 | 54.2 | 64.7 | 72.2 |
| Template (b) | zero-shot | 64.5 | 54.2 | 66.2 | 73.1 |
| | finetune | 64.3 | 55.7 | 64.9 | 72.2 |
| Template (c) | zero-shot | 63.1 | 52.4 | 65.3 | 71.7 |
| | finetune | 64.4 | 55.8 | 65.0 | 72.6 |

(b) Ablation studies on different RL methods.

| Multi-Step Optimization | Single-Step Optimization | Overall | Long | Medium | Short |
|---|---|---|---|---|---|
| | | 64.9 | 53.3 | 65.9 | 75.6 |
| ✓ | | 65.7 | 55.4 | 66.6 | 75.2 |
| ✓ | ✓ | 66.2 | 57.7 | 65.6 | 75.3 |

(c) Ablation studies on different zoom-in video clips.

| Setting | Overall | Long | Medium | Short |
|---|---|---|---|---|
| no zoom-in videos | 60.9 | 52.1 | 61.2 | 69.4 |
| unified zoom-in videos | 61.8 | 51.9 | 62.4 | 71.1 |
| random zoom-in videos | 62.3 | 51.9 | 62.3 | 72.6 |
| adaptive zoom-in videos | 66.2 | 57.7 | 65.6 | 75.3 |

(d) Ablation studies on different numbers of maximum inference iterations.

| #iteration | Overall | Long | Medium | Short |
|---|---|---|---|---|
| 1 | 60.9 | 52.1 | 61.2 | 69.4 |
| 2 | 65.4 | 56.0 | 65.9 | 74.4 |
| 3 | 66.2 | 57.7 | 65.6 | 75.3 |
| 4 | 66.1 | 56.9 | 65.9 | 75.4 |

final response, thereby encouraging timely termination and preventing excessively long chains of thought. Evaluation is conducted with VLMEvalKit (Duan et al., 2024).

**Training Settings.** In Stage 1, we fine-tune the model on 153k video instruction-following samples using a learning rate of $1e^{-5}$ and a batch size of 128. Stage 2 involves further fine-tuning on 38k chain-of-thought (CoT) examples, with the same learning rate and batch size. In the final decoupled reinforcement finetuning stage, we select training samples from CG-Bench (Chen et al., 2025a) and MovieChat (Song et al., 2024) whose rollouts are neither entirely correct nor entirely incorrect for training. This stage uses a learning rate of $1e^{-6}$, a batch size of 32, and performs 8 rollouts per sample.

## 5.2 MAIN RESULTS

To demonstrate the long video understanding ability, we evaluate our model on common benchmarks Video-MME (Fu et al., 2025a), LongVideoBench (Wu et al., 2024), LVBench (Wang et al., 2025b), and MLVU (Zhou et al., 2025). These benchmarks contain videos exceeding one hour, posing great challenges to Video MLLMs. Results are shown in Table 1. **First**, compared with our baseline Qwen2.5-VL (Bai et al., 2025), LOVE-R1 outperforms it by an average of 3.1% points across 4 benchmarks, especially 6.2% points on LVBench. The strong performance shows that our adaptive zoom-in mechanism mitigates the dilemma between sampling density and spatial details faced by the uniform sampling mechanism. **Second**, compared with non-reasoning long video understanding models, which use complex token compression methods or rule-based key frame selection methods, our LOVE-R1 with only 16k context still achieves strong performance on VideoMME (66.2%), LongVideoBench (60.1%) and LVBench (48.2%). And we believe our method can combine with other token compression methods to process more frames and get higher performance. **Further**, compared with open-source reasoning video models, LOVE-R1 outperforms them by a clear margin, demonstrating that simple reasoning can not tackle the long video understanding problem while our multi-step workflow works well on it. We also provide short video understanding benchmark results in Appendix A. And LOVE-R1 still achieves competitive performance compared with other models.

## 5.3 ABLATION STUDIES

**Effect of different video templates.** In Table 2a, we ablate different slow-fast templates. In these experiments, we randomly zoom in on a slow video. Although Templates (a) and (b) perform better in zero-shot evaluation, the performance even degrades after finetuning. We hypothesize it is because these templates are significantly different from the pretraining one. The single video in Template (a) has different resolutions and frame rates, while Template (b) segments the single video into multiple clips. The great template discrepancies can not be mitigated by small-scale finetuning thus degrading the performance. The fast video in Template (c) is still a complete video which is in line with the pretraining one. The model only needs to link slow videos to the corresponding clips

Table 3: Comparisons with reasoning models with fixed frame sampling mechanisms.

| Reasoning Mode | Overall | Long | Medium | Short |
|---|---|---|---|---|
| Single-step reasoning with 128 slow frames | 63.0 | 52.3 | 62.2 | 74.6 |
| Single-step reasoning with 768 fast frames | 62.9 | 55.0 | 64.8 | 68.8 |
| Multi-step reasoning with adaptive zoom-in ability | 66.2 | 57.7 | 65.6 | 75.3 |

within the fast video. Thus, it can be adopted by finetuning on a small scale of data and achieves the highest 64.4% score after finetuning. We use this template as the final template.

**Does decoupled reinforcement finetuning help?** In multi-step reasoning, outcome rewards can not provide fine-grained process supervision, which hinders the zoom-in ability. Therefore, we propose decoupled reinforcement finetuning, which explicitly optimizes zoom-in CoTs by breaking down multi-step reasoning into multiple single-step reasoning. As shown in Table 2b, coupling multi-step and single-step optimization achieves the highest performance 66.2%, outperforming the standard GRPO algorithm by 0.5%.

**Performance comparisons across different training stages.** To effectively build the new model while preserving the pretraining knowledge, we propose a three-stage post-training recipe. Performance in each stage is shown at the bottom of Table 1. After slow-fast template finetuning, the model successfully adapts to the new video template with only 153k data. After the CoT cold start, the model is equipped with the basic adaptive zoom-in ability. The performance on LongVideoBench and LVBench is increased by 4.1% and 1.5% while preserving the performance on other benchmarks, showing the high-quality of our CoT data. Finally, after decoupled reinforcement finetuning, the overall performance is enhanced and the model in Stage 3 achieves the highest performance.

**Is the model really able to zoom in on an informative video clip?** We provide three baselines: 1) no zoom-in videos, in which the model is only provided with the fast video; 2) unified zoom-in videos, in which we sample 32 unified frames from the video as the slow video; 3) random zoom-in videos, in which we randomly select a 30-second clip as the slow video. As shown in Table 2c, compared to baselines, LOVE-R1 adaptively zooms in on clips of interest and achieves the highest performance, demonstrating that the model can truly select the clips related to the questions.

**Effect of different numbers of maximum inference iterations.** In this work, we limited the number of reasoning steps to 3 due to resource constraints. In Table 2d, we study whether more reasoning steps can achieve higher performance. With only one reasoning step, the model can only provide the answer with the fast video and achieves the lowest performance 60.9%. With more reasoning steps, the model can zoom in on some clips of interest and further correct them by sampling other clips when selecting the wrong video clips. This multi-step reasoning paradigm greatly improves the model capacity and achieves higher performance. But the performance is saturated after 3 steps, possibly due to the training context length. We believe this paradigm has great scalability to more reasoning steps.

**Does the improved performance come from reasoning?** We compare LOVE-R1 with single-step reasoning models which can not zoom in on video clips in Table 3. The two baselines are trained with standard single-step GRPO and the same dataset. The model with 128 slow frames gets rich spatial details while missing many temporal clues when tackling long videos. Thus, this model gets high short video performance and low long video results. In contrast, the model with 768 fast frames preserves more temporal clues and gets high performance on long videos but low performance on short videos. Our model with adaptive zoom-in ability achieves a great balance between sampling density and spatial details thus achieving the highest performance. We also note that both baselines underperform Qwen2.5-VL (Bai et al., 2025). We hypothesize the reason is that Video-MME is a dataset for evaluating perception ability rather than reasoning ability (unlike STEM). The same phenomenon can be found between Qwen3-VL-Instruct (Team, 2025) and Qwen3-VL-thinking.

## 5.4 ANALYSIS OF THE NUMBER OF REASONING STEPS ON VIDEOMME

We summarize the statistics of models trained with decoupled reinforcement learning and the original GRPO algorithm on VideoMME in Table 4. Based on these results, we draw the following conclusions: (1) Both models can perform early stopping, as there are many questions with two reasoning steps. (2) Both models can determine whether sufficient information has been gathered, as the accuracy of two-step reasoning is higher than that of three-step reasoning, which means that

Table 4: Analysis of the number of reasoning steps on VideoMME. #Ques and #acc denote the number of questions and the number of questions with correct answers.

| Model | 1 step | | 2 steps | | 3 steps | |
|---|---|---|---|---|---|---|
| | #Ques (rate) | #acc (rate) | #Ques (rate) | #acc (rate) | #Ques (rate) | #acc (rate) |
| Standard GRPO | 0 (0.0%) | 0 (0.0%) | 2592 (96.0%) | 1716 (66.2%) | 106 (3.9%) | 59 (55.7%) |
| Ours | 0 (0.0%) | 0 (0.0%) | 1923 (71.2%) | 1349 (70.2%) | 777 (28.8%) | 438 (56.4%) |

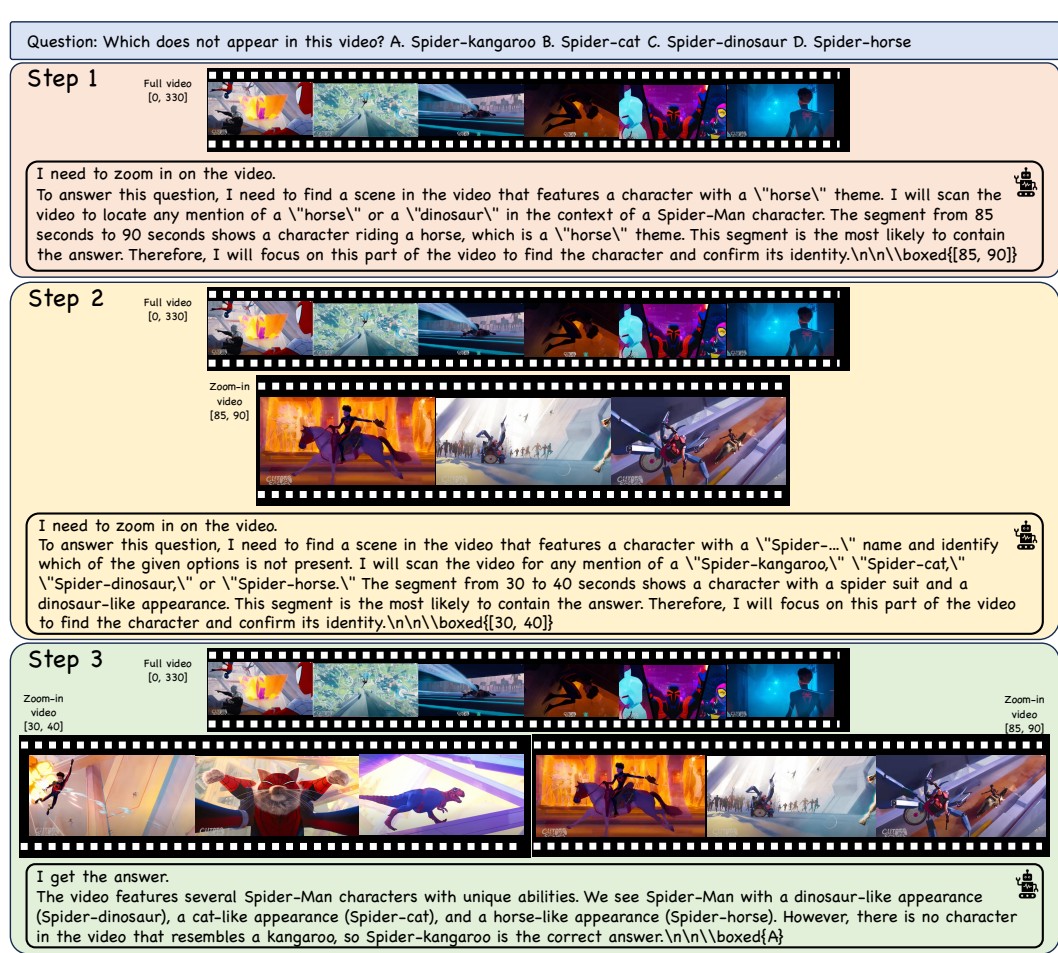

Figure 4: Visualization of gathering multiple clues via multi-step reasoning. The video is taken from Video-MME (vid: edAu5_O4C54).

the model is confident to perform early stopping. (3) Our model training with decoupled reinforcement learning gets a higher two-step reasoning accuracy, showing that our model can zoom in on relevant video clips more accurately. (4) Our model has more reasoning steps, showing that it can determine whether sufficient information has been gathered and conduct more reasoning steps to get sufficient information. These results highlight the advantages of decoupled reinforcement learning in optimizing the zoom-in ability.

## 5.5 VISUALIZATION

We empirically find several promising reasoning patterns after reinforcement learning:

**Multi-hop reasoning.** We find that the model can gather multiple clues from different video clips, as shown in Figure 4. The question asks which does not appear in the video. To answer the question,

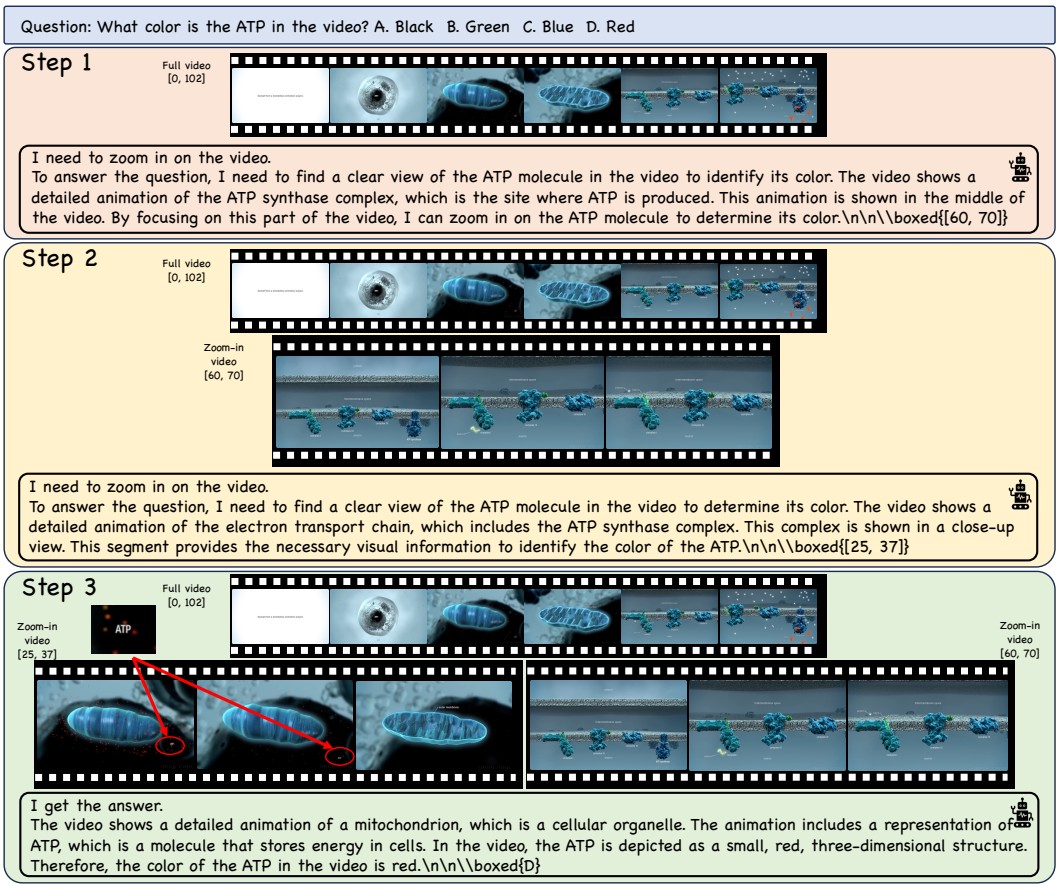

Figure 5: Visualization of self-correction via multi-step reasoning. The video is taken from Video-MME (vid: 39HTpUG1MwQ).

the model in the first step zooms in on the clip from 85 to 90 seconds and finds the Spider-Horse in it. And then, in the second step, the model zooms in on the clip from 30 to 40 seconds to find Spider-Dinosaur and Spider-Cat. Excluding these three options, the model gets the final correct answer A in the third step.

**Self-correction.** Once the model zooms in on an erroneous video clip, it has the ability to find another video clip until the correct one is found. As shown in Figure 5, the model finds a wrong clip in the first step but correctly finds the related video clip in the second step, which contributes to the final correct answer.

More visualizations can be found in Appendix F.

## 6 CONCLUSION

In this work, we propose LOVE-R1, which formulates long video understanding as a multi-step reasoning process. The model with the decision ability, zoom-in ability, and answering ability can adaptively zoom in on a few video clips to get enough spatial details before providing the final answer. This slow-fast adaptive frame sampling mechanism achieves a great trade-off between sampling density and spatial details. To provide fine-grained process rewards, we decouple the multi-step reasoning into multiple single-step reasoning and optimize the internal zoom-in ability explicitly. Experiment results show that our decouple reinforcement finetuning achieves higher performance than the standard GRPO algorithm, which is solely based on outcome rewards and the resulting LOVE-R1 achieves state-of-the-art performance on common long video understanding benchmarks. We hope our work can provide a new paradigm to tackle the long video understanding problem.

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

# APPENDIX

## A    MORE EXPERIMENT RESULTS

Table 5: Evaluation results on short video benchmarks.

| Models | Size | MVBench | Video-MME (short) |
|--------|------|---------|-------------------|
| Video-LLaVA (Lin et al., 2024) | 7B | 41.0 | 45.3 |
| LLaMA-VID (Li et al., 2024d) | 7B | 41.9 | - |
| ShareGPT4Video (Chen et al., 2024a) | 8B | 51.2 | 48.3 |
| LLaVA-NeXT-Video (Zhang et al., 2024b) | 7B | 33.7 | - |
| VideoLLaMA2 (Cheng et al., 2024) | 7B | 54.6 | - |
| VideoChat2 (Li et al., 2024b) | 7B | 60.4 | 48.3 |
| LLaVA-OneVision (Li et al., 2024a) | 7B | 56.7 | - |
| Vamba (Ren et al., 2025) | 10B | 60.4 | - |
| VideoChat-T (Zeng et al., 2025) | 7B | 59.9 | - |
| LongVILA (Chen et al., 2024b) | 7B | 67.1 | 69.0 |
| LongVU (Shen et al., 2024) | 7B | 66.9 | 64.7 |
| Video-R1 (Feng et al., 2025) | 7B | 63.9 | - |
| LongVILA-R1 (Chen et al., 2025b) | 7B | 67.6 | 76.8 |
| LOVE-R1 | 7B | 66.6 | 75.3 |

**Experiment Results on Short Video Understanding Benchmarks.**    In Table 5, we compare LOVE-R1 with other models on common short video understanding benchmarks MVBench (Li et al., 2024b) and the short part of Video-MME (Fu et al., 2025a). Results show that LOVE-R1 also achieves competitive short video understanding performance.

Table 6: Evaluation results on Charades_STA.

| Models | mIoU | R@0.3 | R@0.5 | R@0.7 |
|--------|------|-------|-------|-------|
| Qwen2.5-VL 7B | 44.4 | 65.9 | 46.2 | 24.3 |
| LOVE-R1 | 46.4 | 73.8 | 49.5 | 22.0 |

**Experiment Results on Temporal Grounding Benchmarks.**    In this work, we propose a model with adaptive zoom-in ability that requires basic grounding ability. In Table 2c, we have demonstrated that our LOVE-R1 can truly zoom in on clips of interest, which significantly improves the performance. In Table 6, we further evaluate the performance on a temporal grounding benchmark Charades_STA (Gao et al., 2017). LOVE-R1 achieves a higher mIoU than our base model Qwen2.5-VL, demonstrating its strong temporal grounding ability, especially at low IoU thresholds. We would like to highlight that we do not include any temporal grounding data in the stage 2 and stage 3 of training. And the accuracy reward for zoom-in is a binary reward based on whether the predicted timespan overlaps with the ground truth ones (these are grounded QA samples). And we believe including temporal grounding data can further improve the temporal grounding ability.

## B    COT DATA CONSTRUCTION

### B.1    SOURCE DATA SELECTION

To facilitate the evaluation of the accuracy of the zoom-in video intervals, we select two widely used grounded video question answering datasets NExT-GQA (Xiao et al., 2024) and CG-Bench (Chen et al., 2025a). Each question in these datasets is annotated with related timespans. NExT-GQA is a short video dataset with videos within 3 minutes while CG-Bench is a long video dataset with videos ranging from 10 to 60 minutes. Further, we select some videos ranging from 2 to 3 minutes in LLaVA-Video-178k (Zhang et al., 2024c) and use the pseudo timespans annotated by Wang et al. (2025a). To enhance the diversity of CoT data, in addition to the local question mentioned above, we also select some global questions in FineVideo (Farré et al., 2024) which have some keywords in the question, like "main purpose", "main characters", and "main message".

Figure 6: Our CoT data construction pipeline. To ensure the data quality, we perform strict data pre-processing and post-processing by filtering out low-quality annotations and CoTs. We also use a strong reasoning model, Gemini 2.5 pro, to annotate CoT data, ensuring the content of CoTs is reasonable and high-quality.

## B.2 CoT Data Construction Pipeline

In this work, we decouple the whole pipeline as multiple single-step reasoning, thus we also collect single-step CoTs separately. For each question in NExT-GQA, CG-Bench, and LLaVA-Video-178k, we collect a zoom-in CoT and an answer CoT. For FineVideo, we collect only an answer CoT for each question. The overall pipeline is shown in Figure 6:

---

**The prompt for filtering high-quality timespan annotations**

Question:
{question}

Answer:
{answer}

Do the provided frames contain the visual clues for the answer of the question? Yes or No?

---

**Question Cleaning.** We empirically find that some questions in CG-Bench are extremely hard to answer and the corresponding timespan is inaccurate. We first prompt GPT-4o Hurst et al. (2024) to determine whether the timespan is relevant to the question with the prompt shown above. For questions with correct timespans, we then use Qwen2.5-VL 7B Bai et al. (2025) to filter out questions that can not be answered with ground truth video clips. The resulting data are used for annotation.

---

**The prompt for generating captions for short video clips**

Elaborate on the visual and narrative elements of the video briefly.

---

**Video Captioning.** Since videos in CG-Bench are too long to localize the relevant video clips from the raw videos for recent APIs, we divide the long videos into multiple non-overlapped short clips (10s) and use Qwen2.5-VL 7B Bai et al. (2025) to caption each short video clip.

---

**The prompt for answer CoT construction**

Based on the video and the user question, first provide your reasoning, and then provide the option letter of your final answer within \boxed{}. You can not use audio information during reasoning.

---

---

The prompt for zoom-in CoT construction

You do not know the answer and you should zoom in a specific video segment to answer the question based on your reasoning following the instructions.

## INSTRUCTIONS
- Based on the captions and the user question, first determine what information is needed to answer the question; then, provide your reasoning to localize the video segment that contains the key information and finally, provide the specific video segment within \boxed{[start_time, end_time]}.
- In the reasoning, you should specify how you localize the specific video segment.
- The segment should be presented as [start_time, end_time] in integer seconds. For example, [110, 130].
- You do not know the question answer through the reasoning.
- You can not use audio information during reasoning.

---

**CoT Construction.** To ensure the high-quality of CoTs, we use the Gemini 2.5 pro (Comanici et al., 2025) as the annotator and use different prompts for answer CoTs and zoom-in CoTs as shown above. For short videos, we directly use the raw videos for annotation. For long videos, we use captions as input for zoom-in CoTs while ground truth video clips for answer CoTs. This design can efficiently collect high-quality CoTs under the constraints of APIs.

**Cleaning and Filtering.** Data quality is essential for high-performance models. We perform some rule-based accuracy filtering and format filtering for the collected CoTs. For accuracy filtering, we filter out CoTs with wrong answers or wrong zoom-in timespans (IoU < 0.1). For format filtering, CoTs with repeated patterns and undesired styles are filtered. For example, Gemini 2.5 pro may refer to the words in captions or the voice in videos during reasoning although we prompt it to act as if watching the true video and only focusing on visual information. To ensure consistency, the time representation is standardized to seconds. The resulting 38k CoTs are used for training. An example of our collected CoT is shown in Figure 1.

## C  MORE ANALYSES

### C.1  ERROR ANALYSIS

Although achieving promising results, our model also fails in some cases. We classify the errors into three types: (1) **Modality-Lacking Error**: Questions that cannot be answered using visual information alone; (2) **Localization Error**: Cases where the model either fails to attend to the relevant video clips or only focuses on a subset of them, leading to incorrect reasoning; (3) **Answer Error**: Cases where the model successfully localizes the correct video segments but still generates an incorrect answer.

We randomly collect 20 questions in VideoMME with two reasoning steps and 40 questions with three reasoning steps whose answers are incorrect, and analyze them manually. We find that the two types of questions demonstrate different error patterns:

- Among the questions with two reasoning steps, there are 5 modality-lacking errors (25.0%), 7 localization errors (35.0%), and 8 answer errors (40.0%)

- Among the questions with three reasoning steps, there are 7 modality-lacking errors (17.5%), 25 localization errors (62.5%), and 8 answer errors (20.0%)

Within the localization errors, we identify some common failure cases:

- Multi-Clue Question: The answer is supported by multiple clues and the model only zooms in on a part of them, such as "Which of the following statements is not correct?", counting problems like "How many crystals with cat in boots does Death break" and "In what way is the penultimate goal in the video accomplished?", and sequence order problems like "What is the accurate sequence for the video to present its content?". (15 of 32)

---

- Incoherent video: These include spliced clips, knowledge-based videos, news reports, and documentaries, which often lack a clear, temporally coherent narrative thus making the model easy to fail. (23 of 32)

Based on the statistics above, we can draw the following conclusions: (1) **Our model has the decision-making ability.** The localization error rate for two-step reasoning is significantly lower than the one for three-step reasoning, demonstrating that the model can determine whether the visual information is sufficient to answer the question. If the information is sufficient, the model can finish the reasoning with an early stop. And our model is good at answering questions with a single clue. (2) **Most multi-clue questions belong to three-step reasoning.** We observe that a key source of errors is the constraint of limiting training and inference to at most three reasoning steps, which restricts the model's ability to fully explore and integrate multiple visual clues. In Table 2d, we have demonstrated the scalability of the reasoning steps, and we believe scaling up the training context can further improve performance. (3) **Understanding is crucial for localization.** For incoherent videos, understanding them requires a strong base model. If the model does not understand the video briefly, it is unlikely to localize the correct segment, even with effective reasoning. (4) **These questions are hard samples.** We find that our baseline model Qwen2.5-VL 7B also fails on 73.3% of them. This highlights that the capabilities of the base model are a critical factor influencing the overall performance of our reasoning framework.

## C.2 EFFICIENCY ANALYSIS

Table 7: Efficiency analysis of the zoom-in operation on VideoMME.

| Model | Total Time | Memory | Accuracy |
|---|---|---|---|
| Qwen2.5-VL 7B | 1h25m | 27G | 65.2 |
| LOVE-R1 | 3h27m | 28G | 66.2 |

The primary motivation of our adaptive zoom-in mechanism is to process long videos within acceptable memory as hour-length videos can not be processed by uniform sampling on consumer-grade GPUs. Therefore, we leverage the model's reasoning ability to select relevant video clips, effectively trading computation time for reduced memory usage.

We provide an analysis of inference time efficiency in Table 7. Experiments are conducted on 8 L20 GPUs without using vllm. The total time includes data loading time, video decoding time, and model forward time. As a test-time scaling method, our method with reasoning ability needs around 2.3x inference time with almost the same memory consumption, which is a common phenomenon for all reasoning models.

## D LIMITATION

During the development of LOVE-R1, we find that the performance of recent models is largely constrained by the quality of existing long video understanding training data. Open-sourcing large-scale high-quality long video understanding datasets will make a significant contribution to the community. Further, due to limited computation resources, our model context is limited to 16k. We believe that extending the context length, as demonstrated by LongVILA-R1 (Chen et al., 2025b), allows models to process more frames and conduct more reasoning steps, which can lead to further performance improvements and is left as future work.

## E LLM USAGE

In this work, LLMs make contributions in two aspects:

1. We use LLMs to collect the CoT dataset as mentioned above.
2. We use LLMs to improve paper writing.

# F    MORE VISUALIZATION

We provide more visualizations in Figure 7 and Figure 8. Results show that LOVE-R1 with adaptive zoom-in ability can select clips of interest to zoom in on, thus enhancing the long video understanding ability.

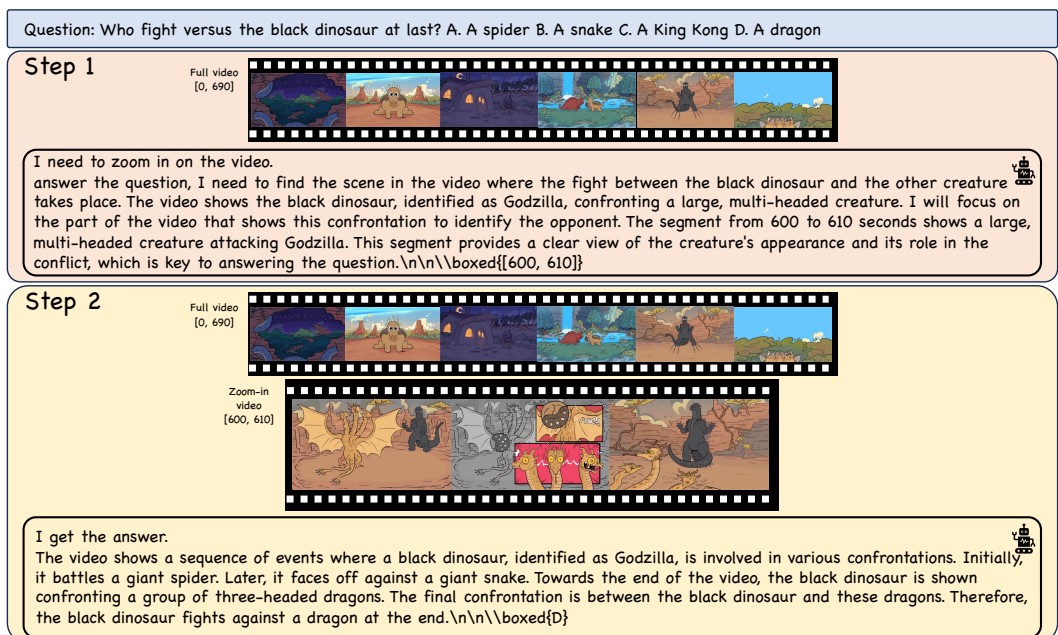

Figure 7: Visualization of LOVE-R1 inference results. The video is taken from Video-MME (vid: -XpJeDGh8No).

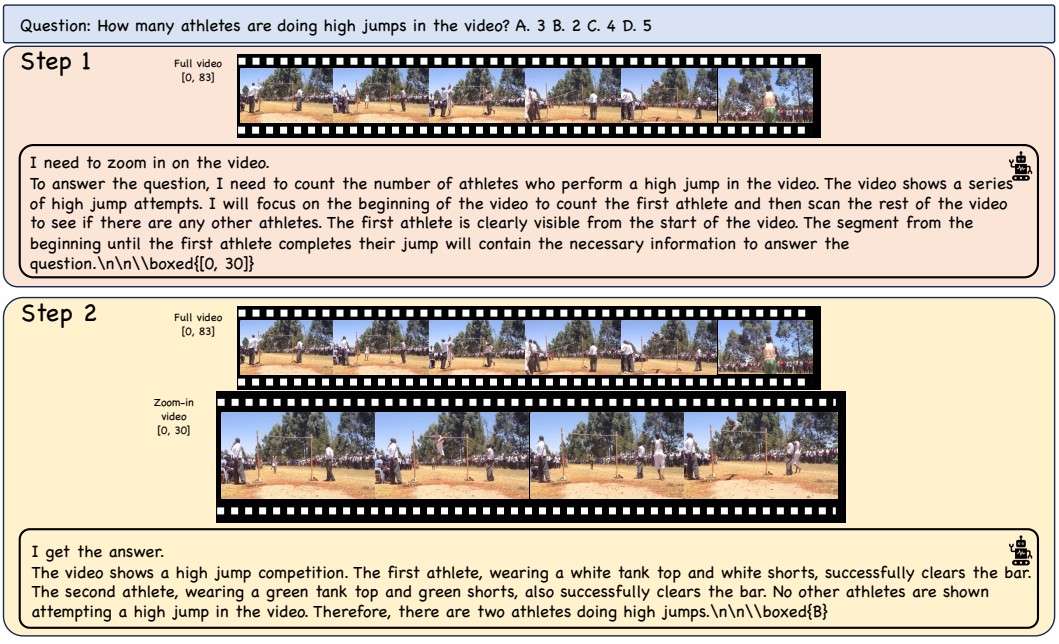

Figure 8: Visualization of LOVE-R1 inference results. The video is taken from Video-MME (vid: -qTAeVGl_e8).

