# OpenReview forum: "LOVE-R1: Advancing Long Video Understanding with Adaptive Zoom-in Mechanism via Multi-Step Reasoning"
_ICLR.cc/2026/Conference — Submitted to ICLR 2026_

### Official Review · Reviewer_yZYH · 2025-10-20

**Soundness:** 3
**Presentation:** 2
**Contribution:** 3
**Rating:** 6
**Confidence:** 3

**Summary:**

The paper proposes LOVE-R1, a large video-language model designed for efficient and accurate long video understanding. The approach combines dense, low-resolution frame sampling for global temporal context and selectively zooms in on high-resolution segments when finer spatial details are necessary, as determined by the model’s reasoning process. The method is trained with a three-stage post-training paradigm. Experiments on several long video benchmarks show consistent improvements over strong baselines such as Qwen2.5-VL, achieving state-of-the-art performance with a 3.1% average gain. Overall, the paper contributes a novel adaptive sampling framework and a reasoning-based training pipeline that effectively balance temporal coverage and spatial detail for long video understanding.

**Strengths:**

**1. Originality:**
The paper addresses a fundamental limitation of current video-language models, the trade-off between temporal coverage and spatial fidelity under constrained context length. By introducing an adaptive, query-driven zoom-in mechanism, it reframes long video understanding as a progressive multi-step reasoning process (decision, zoom-in, answer). This formulation is both intuitive and impactful, demonstrating clear conceptual originality.

**2. Technical Soundness:**
The proposed three-stage post-training framework, comprising slow–fast template finetuning, chain-of-thought (CoT) initialization, and decoupled reinforcement optimization, is logically structured and empirically validated. Each stage contributes a distinct function.

**3. Empirical Validation:**
The experimental evaluation is extensive and convincing. The proposed LOVE-R1 model consistently outperforms strong baseline across multiple long video understanding benchmarks (Video-MME, LongVideoBench, LVBench, MLVU). The ablation studies (Tables 2–3) clearly attribute gains to specific model components, confirming the utility of adaptive sampling and multi-step reasoning.

**Weaknesses:**

**1. Marginal Gain from Decoupled RL:**
A core innovation presented for the training process is the "decoupled reinforcement finetuning" (Stage 3). The ablation study in Table 2b shows that this decoupled optimization (labeled "Single-Step Optimization") achieves an overall score of 66.2%, only a marginal 0.5% improvement over the standard multi-step GRPO baseline ("Multi-Step Optimization"), which scored 65.7%. This minimal gain does not strongly justify the added complexity of the decoupled approach, and the paper does not clearly demonstrate the specific conditions or failure cases under which this method becomes essential.

**2. Missing Error Analysis and Efficiency Metrics:**
The paper lacks a crucial discussion of the model's specific failure modes. While ablation studies compare against baselines like "random zoom-in", there is no qualitative or quantitative analysis of when the LOVE-R1 model itself fails. For instance, the analysis does not explore the frequency or nature of incorrect zoom-in localizations, errors in the decision-making step (i.e., deciding to zoom vs. to answer), or misinterpretations of the detailed "slow video" frames.

**3. Unknown Computation Cost:**
Furthermore, the paper omits any reporting on computational cost. A primary motivation for an adaptive "slow-fast" mechanism is to balance performance with efficiency. However, the paper provides no metrics on inference time, memory footprint, or a clear discussion of the practical efficiency trade-offs between sampling density and spatial resolution.

**Questions:**

1. Given the marginal 0.5% gain from decoupled reinforcement finetuning, under what conditions does this method significantly outperform standard GRPO, justifying its added complexity?
2. Can the authors provide error analysis on zoom-in failures, such as mislocalization rates or incorrect decision-to-zoom vs. answer choices?

---

> ### Author Response · Authors · 2025-11-22
> **Response to Reviewer Je7w (Part 1 of 3)**
>
> We sincerely appreciate the reviewer's constructive comments, which have helped us improve the quality of the manuscript. We address the reviewer's comments as follows:
>
> > **W1**: Marginal Gain from Decoupled RL
>
> **RE**:
>
> | Multi-Step Optimization | Single-Step Optimization | MLVU | VideoMME | LongVideoBench | LVBench | Avg. gains |
> | :---: | :---: | :---: | :---: | :---: | :---: | :---: |
> | |  | 66.7 | 64.9 | 59.7 | 46.2 | - |
> | ✅ | | 67.5 | 65.7 | 59.2 | 46.5 | +0.35 |
> | ✅ | ✅ | 67.4 | 66.2 | 60.1 | 48.2 | +1.10 |
>
> **First of all, training with decoupled reinforcement finetuning achieves stable improvements across all benchmarks.** In the table above, we provide more evaluation results on all benchmarks. With the original GRPO algorithm (multi-step optimization), the performance only improves by an average of 0.35 points, and even slightly degrades on LongVideoBench. Combined with single-step optimization, the improvements are more stable and significant with an average of 1.10 points, demonstrating the effectiveness of the single-step optimization.
>
> | Model | 1 step |  1 step | 2 steps |  2 steps | 3 steps |  3 steps |
> | :--- | :---: | :---: |  :---: | :---: | :---: | :---: |
> | |\#Ques (rate)  |  \#acc (rate)  |  \#Ques (rate)  |  \#acc (rate)  |  \#Ques (rate)  |  \#acc (rate) |
> | Standard GRPO | 0 (0.0%)  |  0 (0.0%) | 2592 (96.0%) | 1716 (66.2%) | 106 (3.9%) |  59 (55.7%)|
> | Ours | 0 (0.0%) | 0 (0.0%) | 1923 (71.2%) | 1349 (70.2%) | 777 (28.8%) | 438 (56.4%) |
>
> **Second, training with decoupled reinforcement finetuning achieves higher accuracy when reasoning with multiple steps.** We summarize the reasoning step statistics of models training with decoupled reinforcement learning and the original GRPO algorithm on VideoMME in the table above. #Ques and #acc denotes the number of questions and the number of questions with correct answers. We find two crucial characteristics of the model training with decoupled reinforcement finetuning:
> - Higher accuracy on questions with two reasoning steps. Training with decoupled reinforcement finetuning, the model can directly find the corresponding video clips for single-clue questions with a higher accuracy. Therefore, the accuracy of questions with two reasoning steps is higher.
> - More reasoning steps. The model training with decoupled reinforcement finetuning can determine whether key clues are gathered and zooms in on relevant clips if key clues are missing.
>
> **Finally, training with decoupled reinforcement finetuning will not increase training or inference complexity.** During training, we use some data with ground truth timestamps for the single-step optimization and other data for the multi-step optimization. The former data will only be used for the single-step optimization. If we do not apply the single-step optimization, these data will also be used for the multi-step optimization. Therefore, the training time cost is the same. And models with different optimization algorithms adopt the same inference mechanism, thus the inference time cost is also similar.

---

> ### Author Response · Authors · 2025-11-22
> **Response to Reviewer Je7w (Part 2 of 3)**
>
> > **W2**: Missing Error Analysis
>
> **RE**: We classify the errors into three types:
> - **Modality-Lacking Error**: Questions that cannot be answered using visual information alone;
> - **Localization Error**: Cases where the model either fails to attend to the relevant video clips or only focuses on a subset of them, leading to incorrect reasoning;
> - **Answer Error**: Cases where the model successfully localizes the correct video segments but still generates an incorrect answer.
>
> We randomly collect 20 questions in VideoMME with two reasoning steps and 40 questions with three reasoning steps whose answers are incorrect, and analyze them manually. We find that the two types of question demonstrate different error patterns:
> - Among the questions with two reasoning steps, there are 5 modality-lacking errors (25.0%), 7 localization errors (35.0%), and 8 answer errors (40.0%).
> - Among the questions with three reasoning steps, there are 7 modality-lacking errors (17.5%), 25 localization errors (62.5%), and 8 answer errors (20.0%).
>
> Within the localization errors, we identify some common failure cases:
> - Multi-Clue Question: The answer is supported by multiple clues and the model only zooms in on a part of them, such as "Which of the following statements is not correct?", counting problems like "How many crystals with cat in boots does Death break" and "In what way is the penultimate goal in the video accomplished?", and sequence order problems like "What is the accurate sequence for the video to present its content?". (15 of 32)
> - Incoherent video: These include spliced clips, knowledge-based videos, news reports, and documentaries, which often lack a clear, temporally coherent narrative thus making the model easy to fail. (23 of 32)
>
> Based on the statistics above, we can draw the following conclusions:
> - **Our model has the decision-making ability.** The localization error rate for two-step reasoning is significantly lower than the one for three-step reasoning, demonstrating that the model can determine whether the visual information is sufficient to answer the question. If the information is sufficient, the model can finish the reasoning with an early stop. And our model is good at answering questions with a single clue.
> - **Most multi-clue questions belong to three-step reasoning.** We observe that a key source of errors is the constraint of limiting training and inference to at most three reasoning steps, which restricts the model’s ability to fully explore and integrate multiple visual clues. In Table 2(d), we have demonstrated the scalability of the reasoning steps, and we believe scaling up the training context can further improve performance. Unfortunately, we do not have sufficient resources to extend the context.
> - **Understanding is crucial for localization.** For incoherent videos, understanding them requires a strong base model. If the model does not understand the video briefly, it is unlikely to localize the correct segment, even with effective reasoning.
> - **These questions are hard samples.** We find that our baseline model Qwen2.5-VL 7B also fails on 73.3% of them. This highlights that the capabilities of the base model are a critical factor influencing the overall performance of our reasoning framework.
>
> **These results are included in Appendix C.1 of our revised manuscript.**

---

> ### Author Response · Authors · 2025-11-22
> **Response to Reviewer Je7w (Part 3 of 3)**
>
> > **W3**: Unknown Computation Cost
>
> **RE**:
>
> | Model | Total Time | Memory | Accuracy |
> | :--- | :---: | :---: | :---: |
> | Qwen2.5-VL 7B | 1h25m | 27G | 65.2 |
> | LOVE-R1 | 3h27m | 28G | 66.2 |
>
> The primary motivation of our adaptive zoom-in mechanism is to process long videos within acceptable memory as hour-length videos can not be processed by uniform sampling on consumer-grade GPUs. Therefore, we leverage the model's reasoning ability to select relevant video clips, effectively trading computation time for reduced memory usage.
>
> We provide an analysis of inference time efficiency on VideoMME in the table above. Experiments are conducted on 8 L20 GPUs without using vllm. The total time includes data loading time, video decoding time, and model forward time. As a test-time scaling method, our method with reasoning ability needs around 2.3x inference time with almost the same memory consumption, which is a common phenomenon for all reasoning models.
>
> **These results are included in Appendix C.2 of our revised manuscript.**
>
> :
> > **Q1**: Given the marginal 0.5% gain from decoupled reinforcement finetuning, under what conditions does this method significantly outperform standard GRPO, justifying its added complexity?
>
> **RE** Please see W1.
>
> :
> > **Q2**: Can the authors provide error analysis on zoom-in failures, such as mislocalization rates or incorrect decision-to-zoom vs. answer choices?
>
> **RE** Please see W2.

---

### Official Review · Reviewer_Je7w · 2025-10-28

**Soundness:** 3
**Presentation:** 3
**Contribution:** 3
**Rating:** 6
**Confidence:** 5

**Summary:**

This paper proposes LOVE-R1, a VLM with an adaptive zoom-in mechanism based on multi-step reasoning. The mode processes long videos with a "slow-fast" frame sampling template for fine-grained spatial details in key clips. With a three-stage pipeline: (1) slow-fast template fine-tuning, (2) Chain-of-Thought (CoT) cold start with annotated data to build basic reasoning abilities , (3) decoupled reinforcement fine-tuning to address the sparsity of outcome-based rewards. Experiments on long video benchmarks show LOVE-R1 outperforms Qwen2.5-VL.

**Strengths:**

1. The explored problem is meaningful and the motivation is clear. The grounding guided slow frame zoom in makes sense in long video understanding.
2. The slow-fast template design and the three stage training pipeline makes sense. The single-step supervision is crucial in video reasoning.
3. The CoT data construction is meaningful and is a contribution to the community.

**Weaknesses:**

1. The context length adopted in the paper is limited and the scalability remains doubtful. More experiments on models with large context window is necessary.
2. The CoT data derived from grounding data is narrow in domain and fails to cover general long video scenarios, which limits the generalization ability.
3. The paper lacks the efficiency analysis on the computation overhead introduced by the zoom in operation.

**Questions:**

Except the grounding data, are there other types of data that can be used for single-step optimization?

---

> ### Author Response · Authors · 2025-11-22
> **Response to Reviewer Je7w**
>
> We sincerely appreciate the reviewer's constructive comments, which have helped us improve the quality of the manuscript. We address the reviewer's comments as follows:
>
> > **W1**: The context length adopted in the paper is limited and the scalability remains doubtful.
>
> **RE:** In Table 2(d), we have demonstrated that our method enjoys the scalability of the reasoning steps. And conducting more reasoning steps can improve the performance. Unfortunately, we can not provide the results of more reasoning steps due to limited resources.
>
> :
> > **W2**: The CoT data derived from grounding data is narrow in domain and fails to cover general long video scenarios, which limits the generalization ability.
>
> **RE:** As shown in Section B.1, we construct the CoT data from NExT-GQA, CG-Bench, LLaVA-Video-178k, and FineVideo. Among these, LLaVA-Video-178k and FineVideo are widely used video instruction tuning datasets that cover a diverse range of question types. In FineVideo, we specifically select a subset of global questions to complement the local questions from the other sources. The variety in video content and question types enhances the model’s generalization ability.
>
> In our early exploration, we also experimented with incorporating CoT data from LongVILA-R1. However, performance degraded when adding these data, likely due to mismatches in video templates and differences in reasoning patterns.
>
> :
> > **W3:** The paper lacks the efficiency analysis on the computation overhead introduced by the zoom in operation.
>
> **RE**:
>
> | Model | Total Time | Memory | Accuracy |
> | :--- | :---: | :---: | :---: |
> | Qwen2.5-VL 7B | 1h25m | 27G | 65.2 |
> | LOVE-R1 | 3h27m | 28G | 66.2 |
>
> We provide an analysis of the inference time efficiency on VideoMME in the table above. Experiments are conducted on 8 L20 GPUs without using vllm. The total time includes data loading time, video decoding time, and model forward time. As a test-time scaling method, our method with reasoning ability needs around 2.3x inference time with almost the same memory consumption, which is a common phenomenon for all reasoning models. **These results are included in Appendix C.2 of our revised manuscript.**
>
> :
> > **Q1**: Except the grounding data, are there other types of data that can be used for single-step optimization?
>
> **RE**: Without ground truth timespans, we can not provide process rewards. As a substitute, we can use frontier MLLMs as annotators to annotate pseudo timespans for general QAs, which may increase the data diversity.

---

### Official Review · Reviewer_cdEK · 2025-10-29

**Soundness:** 3
**Presentation:** 3
**Contribution:** 2
**Rating:** 2
**Confidence:** 4

**Summary:**

The paper proposes a new reasoning pipeline for Video-LLMs that performs temporal retrieval from the original clip and then zooms in to extract fine-grained details in later passes. The training is divided into three stages, initialized from Qwen2.5-VL. Experiments demonstrate clear gains over the baseline.

**Strengths:**

1. The method is well-explained, and the implementation details are sufficient for the community to reproduce the results.

2. The concept of temporal retrieval followed by multi-pass reasoning is intuitive and well-founded. It makes sense that video reasoning would benefit from zooming into relevant segments after a coarse-level pass.

**Weaknesses:**

1. The concatenation design highlighted in Figure 2 appears straightforward and not technically novel. Its inclusion as a major methodological highlight seems unnecessary.

2. The paper does not evaluate temporal grounding, even though many benchmarks (e.g., Charades-STA) provide standardized evaluation protocols for video temporal localization. Only video-QA experiments are reported, and the results are not state-of-the-art compared to 7B-scale counterparts, despite the proposed model processing more frames and multiple reasoning iterations.

**Questions:**

1. For each retrieval step, is only one slow-motion clip retrieved? If so, how does the model handle cases where clues are distributed across multiple, temporally distant clips? Does the framework support multi-hop retrieval?

2. In Table 1, the Stage 2 results appear to degrade compared to Stage 1 on the first two benchmarks where slow videos are uniformly sampled. This is counterintuitive. Does it suggest that those benchmarks may not actually benefit from detailed subclip reasoning?

3. Does the paper analyze the average number of reasoning iterations performed across different benchmarks? Additionally, can the model adaptively terminate early when sufficient information has been gathered, i.e., does it learn an implicit “early stop” or “reasoning drop-out” behavior?

---

> ### Author Response · Authors · 2025-11-22
> **Response to Reviewer cdEK (Part 1 of 2)**
>
> > **W1**: The concatenation design appears straightforward.
>
> **RE:** In this work, our main contribution is a novel video processing mechanism, which we formulate as a multi-step reasoning problem. The video template, as a key design factor, plays a critical role in shaping both the algorithm’s design and implementation. We compare the proposed appended video template with two alternative variants and demonstrate that it aligns better with the pretraining setup and gets higher performance.
>
> :
> > **W2**: The paper does not evaluate temporal grounding (e.g., Charades-STA).
>
> | Models | mIoU | R\@0.3 | R\@0.5 | R\@0.7 |
> | :--- | :---: | :---: | :---: | :---: |
> | Qwen2.5-VL | 44.4 | 65.9 | 46.2 | 24.3 |
> | LOVE-R1 | 46.4 | 73.8 | 49.5 | 22.0 |
>
> **RE:** We provide the results on Charades\_STA in the table above. LOVE-R1 achieves a higher mIoU than our base model Qwen2.5-VL, demonstrating its strong temporal grounding ability, especially at low IoU thresholds. **These results are included in Appendix A of our revised manuscript.**
>
> There are many **task discrepancies** between long video understanding and temporal grounding, the former aims to answer the questions accurately while the latter aims to localize the boundary completely. We would like to highlight that we do not include any temporal grounding data in the stage 2 and stage 3 of training. And the accuracy reward for zoom-in is a binary reward based on whether the predicted timespan overlaps with the ground truth ones. (these are grounded QA samples.) And we believe including temporal grounding data can further improve the temporal grounding ability.
>
> :
> > **W3**: The results are not state-of-the-art compared to 7B-scale counterparts.
>
> **RE:** We would like to highlight that our LOVE-R1 model outperforms our baseline Qwen2.5-VL by an average of 3.1% points across 4 benchmarks and other reviewers (Reviewer R1ym, Reviewer yZYH) agree with us that our improvements are solid and convincing. Although our method achieves comparable or slightly lower performance than several 7B counterparts on one of the four benchmarks, this can be attributed to two main factors:
> - **Constrained context length**. Due to resource constraints, our model is trained with a 16k context. In contrast, VILAMP and Hour-LLaVA trains with a 32k context and LongVITA-128k trains with a 128k context. Further, VILAMP and Hour-LLaVA process the video at 1fps, which is up to 3600 frames for a one-hour video and is much more than our 768 processed frames. Importantly, in Table 2 (d), we show that our model enjoys great scalability. With a longer context, the model can conduct more reasoning steps and achieve higher performance.
> - **Relatively low baseline**. In our work, we use the widely adopted Qwen2.5-VL as the baseline, which adopts a uniform frame sampling mechanism without token compression and whose performance is lower than other specialist models on long video understanding. We emphasize that our proposed zoom-in mechanism is orthogonal to other token compression methods. The token compression methods can be applied to our fast video to process more frames within a limited context.
>
> :
> > **Q1**: For each retrieval step, is only one slow-motion clip retrieved? If so, how does the model handle cases where clues are distributed across multiple, temporally distant clips? Does the framework support multi-hop retrieval?
>
> **RE:** Yes, the model supports multi-hop retrieval by reasoning with multiple steps, despite the model zooming in on only one clip at each step. As shown in Figure 4, the model in the first step zooms in on the clip from 85 to 90 seconds and finds the Spider-Horse in it. And then, in the second step, the model zooms in on the clip from 30 to 40 seconds to find Spider-Dinosaur and Spider-Cat. Excluding these three options, the model gets the final correct answer A in the third step. This is a typical case for multi-hop retrieval.
>
> Moreover, since the fast video retains the full semantic content of the original video, the model can directly extract multiple clues from it. As demonstrated in Table 2(c), the model achieves a score of 60.9 on VideoMME using only the fast video, confirming its ability to capture some information without fine-grained temporal zooming.
>
> :
> > **Q2**: In Table 1, the Stage 2 results appear to degrade compared to Stage 1 on the first two benchmarks where slow videos are uniformly sampled. This is counterintuitive. Does it suggest that those benchmarks may not actually benefit from detailed subclip reasoning?
>
> **RE:** Not exactly. After cold start, the model only gains the basic reasoning ability. The reasoning ability is constrained by the imitation learning of supervised training and relatively small (38k) and limited data. As a result, the model's performance slightly degrades after cold start. However, after reinforcement finetuning, the reasoning capability is significantly enhanced, leading to improved performance across all benchmarks.

---

> ### Author Response · Authors · 2025-11-22
> **Response to Reviewer cdEK (Part 2 of 2)**
>
> > **Q3**: Does the paper analyze the average number of reasoning iterations performed across different benchmarks? Additionally, can the model adaptively terminate early when sufficient information has been gathered, i.e., does it learn an implicit “early stop” or “reasoning drop-out” behavior?
>
> **RE**:
>
> | Model | 1 step |  1 step | 2 steps |  2 steps | 3 steps |  3 steps |
> | :--- | :---: | :---: |  :---: | :---: | :---: | :---: |
> | |\#Ques (rate)  |  \#acc (rate)  |  \#Ques (rate)  |  \#acc (rate)  |  \#Ques (rate)  |  \#acc (rate) |
> | Standard GRPO | 0 (0.0%)  |  0 (0.0%) | 2592 (96.0%) | 1716 (66.2%) | 106 (3.9%) |  59 (55.7%)|
> | Ours | 0 (0.0%) | 0 (0.0%) | 1923 (71.2%) | 1349 (70.2%) | 777 (28.8%) | 438 (56.4%) |
>
> We summarize the reasoning step statistics of models training with decoupled reinforcement learning and the original GRPO algorithm on VideoMME in the table above. \#Ques and \#acc denotes the number of questions and the number of questions with correct answers. Based on these results, we draw the following conclusions:
> - Both models can perform an early stopping as there are many questions with two reasoning steps.
> - Both models can determine whether sufficient information has been gathered as the accuracy of two-step reasoning is higher than the one of three-step reasoning, which means that the model is confident to perform early stopping.
> - Our model training with decoupled reinforcement learning gets a higher two-step reasoning accuracy, showing that our model can zoom in on relevant video clips more accurately.
> - Our model has more reasoning steps, showing that it can determine whether sufficient information has been gathered and conduct more reasoning steps to get sufficient information.
>
> **These results are included in Section 5.4 of our revised manuscript.**
>
> We sincerely appreciate the reviewer's constructive comments, which have helped us improve the quality of the manuscript.

---

### Official Review · Reviewer_R1ym · 2025-10-31

**Soundness:** 4
**Presentation:** 3
**Contribution:** 3
**Rating:** 6
**Confidence:** 5

**Summary:**

This study proposes the LOVE-R1 model, aimed at addressing the common spatio-temporal trade-off problem in long video understanding. The model constructs long video understanding as a multi-step reasoning process, first obtaining a global temporal overview through a low-resolution, high-density "fast video" stream, and then adaptively "zooming in" on specific key segments as needed, using a high-resolution "slow video" stream to capture spatial details. The model's "zooming in" decision-making ability is optimized through a three-stage training strategy and providing fine-grained process rewards, not solely relying on the correctness of the final answer.

**Strengths:**

The main advantage of love-r1 lies in its novel "amplification" mechanism, which provides a reasonable and effective paradigm for addressing the conflict between temporal density and spatial details in long videos. Second, the proposed "decoupled reinforcement fine-tuning" method tackles the issue of sparse multi-step rewards in GRPO by introducing process rewards, thereby more accurately enhancing the model's temporal localization capability. Finally, the experiments in the paper are solid, achieving performance better than baseline models on multiple long video benchmarks, and demonstrating the effectiveness of its key components (such as the amplification mechanism, video templates, and training method) through comprehensive ablation studies (e.g., Table 2).

**Weaknesses:**

1. The success of the entire reasoning chain depends on the model's ability to correctly identify the area that needs "zooming in" in the first step, but the "fast video" resolution is only around 168x168. If key clues, such as distant small objects or blurred text, cannot be recognized at this resolution, the model may make an incorrect "zooming in" decision in the first step, leading to all subsequent high-resolution analyses being based on wrong segments.
2. The context length may be limited by the "additional video template" (Template c): this template is cumulative, requiring the model to always retain the complete low-resolution "fast video" in the context while appending new high-resolution "slow video" segments at each inference step. This strategy leads to increased token consumption, hindering the model's ability to perform longer and deeper reasoning within a fixed context length, such as 16k.

**Questions:**

1. Will the constructed 38k CoT dataset be open-sourced?

**Details Of Ethics Concerns:**

The repository link in the paper does not comply with the anonymity guidelines.

---

> ### Author Response · Authors · 2025-11-22
> **Response to Reviewer R1ym**
>
> We sincerely appreciate the reviewer's constructive comments, which have helped us improve the quality of the manuscript. We address the reviewer's comments as follows:
>
> > W1: The resolution of fast video is low and distant small objects or blurred text cannot be recognized at this resolution.
>
> **RE1:** The primary goal of the "fast video" is to enable the model to localize relevant video clips within a long video. For example, as shown in Figure 1, the model cannot read the text in the frames of the "fast video" due to the low resolution. Instead, it identifies the relevant clip "adding garlic" around the 23-second mark. The model then zooms in on this clip to clearly read the guidelines on the screen "GARLIC 4 CLOVES, MINCED". Scenarios where key clues like distant small objects or blurred text cannot be recognized under a low resolution are a key advantage of our proposed zoom-in mechanism compared with the uniform sampling mechanism. If the model can already obtain sufficient visual information directly from the "fast video", it may output the solution without zoom-in. Notably, the number of frames and the frame resolution are hyperparameters and can be tuned for an optimal combination.
>
> :
> > W2: The cumulative template leads to increased token consumption, hindering the model's ability to perform longer and deeper reasoning.
>
> **RE2:** During model development, we find that the key to the success of the zoom-in mechanism lies in the dense-sampled "fast video," which provides a global overview of the long video. Retaining the full low-resolution "fast video" in context enables two critical capabilities:
> - **Self-correction**: When the model zooms in on an incorrect video clip, it can revisit the "fast video" and select a different segment until a correct clip is identified. (Please see Figure 5.)
> - **Multi-hop reasoning**: For questions requiring multiple visual clues, the model should also revisit the "fast video" multiple times to locate distinct relevant segments. For instance, in Figure 4, the model first identifies the "Spider-Horse" in the clip from 85 to 90 seconds, then locates "Spider-Dinosaur" and "Spider-Cat" in the 30–40 second segment. By eliminating these three options, it arrives at the correct answer A, in a third step.
>
> As a test-time scaling strategy, the increased token consumption is normal and acceptable.
>
> :
> > Q1: Will the constructed 38k CoT dataset be open-sourced?
>
> **RE3:** Absolutely, we will open-source all assets.
>
> :
> > Concern: The repository link in the paper does not comply with the anonymity guidelines.
>
> **RE4:** In our submitted manuscript, we have not included any links to our repositories. In the references, we cite three prior works for acknowledgment. Since these works have not been formally published, we reference their public webpages. These resources are publicly accessible and do not contain any supplementary material that extends the content of our manuscript or reveals author identities.

---

### Author Response · Authors · 2025-12-02
**Final Response to Reviewers and Summary of Revisions**

We sincerely thank all the reviewers for their thoughtful and constructive feedback. We are encouraged by the recognition of several key aspects of our work:
- We are addressing a fundamental and meaningful limitation in current video-language models (Reviewers `Je7w`, `yZYH`);
- Our motivation is intuitive and reasonable, reflecting conceptual originality (Reviewers `R1ym`, `cdEK`, `Je7w`, `yZYH`);
- The proposed decoupled reinforcement finetuning effectively addresses the issue of sparse multi-step rewards in GRPO and plays a crucial role in accurate video localization (Reviewers `R1ym`, `Je7w`);
- The extensive and rigorous experiments convincingly demonstrate the effectiveness of our approach (Reviewers `R1ym`, `yZYH`).

And we will open-source all assets to support future exploration.

During the discussion phase, we conducted deeper analyses that further revealed some promising properties of our method:
- **Advanced reasoning patterns**: Our multi-step reasoning framework, equipped with a cumulative template, naturally supports self-correction and multi-hop reasoning. Self-correction allows the model to refine its predictions for more precise clip localization, while multi-hop reasoning enables effective retrieval across disjoint video segments. (Reviewer `R1ym` and Reviewer `cdEK`)
- **Early-stop mechanism**: The model learns to assess whether sufficient visual evidence has been gathered to answer the question and can terminate reasoning early when adequate information is available. (Reviewer `cdEK`)
- **Minimal GPU memory overhead**: The primary motivation of our adaptive zoom-in mechanism is to process long videos within acceptable memory. And we find that it enables the processing of more informative video content with nearly the same GPU memory cost as uniform sampling, at the expense of acceptable computational time. (Reviewer `Je7w` and Reviewer `yZYH`)
- **Improved temporal grounding ability**: We demonstrate that our model also outperforms our baseline model on temporal grounding, and the ability is further amplified by the decoupled reinforcement learning. (Reviewer `cdEK`)

These insights, along with analyses on the number of reasoning steps, error types, and efficiency, are now included in the revised manuscript.


We also acknowledge that the model still faces challenges on complex, multi-clue questions and incoherent videos. These questions are also challenging for our base model. We believe that increasing the number of reasoning steps and improving the perception ability of the base model can mitigate the problem.

With these detailed analyses, we hope to adequately address the concerns raised by Reviewer `cdEK`. Specifically:
1. We emphasize the importance of the video template in guiding multi-step reasoning to address W1;
2. We include additional evaluations on temporal grounding to further demonstrate the model’s strong localization capability, addressing W2;
3. We provide a discussion on how token compression methods can be combined with our approach, responding to W3;
4. We highlight a concrete example of multi-hop reasoning in the manuscript to support our claim in Q1;
5. We clarify the distinct roles of supervised fine-tuning (SFT) and reinforcement learning (RL) in the training pipeline to address Q2;
6. We present a thorough analysis of the number of reasoning steps, addressing Q3.

These additions not only strengthen our response but also offer deeper insights into the model’s behavior and capabilities. Further, we are grateful that three other reviewers (`R1ym`, `Je7w`, and `yZYH`) found our improvements to be solid and convincing, and we appreciate their positive feedback.

Once again, we deeply appreciate the time and effort invested by the reviewers and the Area Chair. Your valuable feedback has significantly strengthened the quality and clarity of our work.

---

### Meta-Review · Area_Chair_7PzZ · 2026-01-05

**Summary:**

After reading the paper and the rebuttal. The AC agrees with the concerns raised by the reviewer, including the insufficient method validation caused by the limited context length (R1ym, cdEK, Je7w) and insignificant improvement (cdEK, yZYH), so the AC tends to reject this paper.

**Reviewer Concerns:**

The authors replied to most of the concerns in the rebuttal. However, for the most critical one, the context length, the authors failed to provide strong results.

**Reviewer Scores:**

Reviewer R1ym, cdEK, Je7w mentioned the context length problem, while the author failed to provide the results. So the score may not change, as this is a critical point to validate the proposed method.

---

### Decision · Program_Chairs · 2026-01-26

Reject